# Psychological Impact of Parkinson Disease Delusions on Spouse Caregivers: A Qualitative Study

**DOI:** 10.3390/brainsci11070871

**Published:** 2021-06-29

**Authors:** Caroline J. Deutsch, Noelle Robertson, Janis M. Miyasaki

**Affiliations:** 1Complex Neurology Symptoms Clinic, 1H1.26 WMC, University of Alberta Hospital, Edmonton, AB T6G 2B7, Canada; caroline.deutsch@albertahealthservices.ca; 2Department of Neuroscience, Psychology and Behaviour, University of Leicester, Center for Medicine, Lancaster Road, Leicester LE1 7HA, UK; nr6@leicester.ac.uk; 3Department of Medicine, Parkinson and Movement Disorders Program, University of Alberta, 7-133 Clinical Sciences Building, 11350-83 Avenue, Edmonton, AB T6G 2G3, Canada

**Keywords:** delusions, Parkinson’s disease, psychosis, spouse caregiver

## Abstract

There is growing research on carers of people with Parkinson’s disease (PD) experiences. However, the impact on carers by PD delusions is not specifically examined. We conducted a qualitative study using semi-structured interviews of spouse carers of PD patients with delusions. Thematic analysis was employed using MAXQDA 2018. Twelve spouse participants (SPs) were interviewed. Four themes emerged: Managing incredulity: trying to make sense of delusion content; Hypervigilance: constant alertness to bizarre and threatening discourse and behavior; Defensive strategizing: anticipating delusions and potential consequences; Concealing and exposing: ambivalence about disclosing the effect of delusions yet wanting support. SPs reported effects on their emotional well-being and marital relationship and challenges to an orderly, predictable life. SPs were reluctant to share their experiences due to delusion content (often infidelity and sexual in nature) and a desire to protect their spouses’ image. SPs’ awareness of the potential for delusional thought was low prior to their occurrence. Conclusions: education surrounding potential neurobehavioral changes should occur for patients and carers. Clinicians should be aware that the impact of delusions on carers is often greater than disclosed in clinical interviews. Interdisciplinary teams speaking separately to spousal carers may improve disclosure and delivery of appropriate psychological support.

## 1. Introduction

Delusions are fixed beliefs not based on reality and are a recognized complication of Parkinson’s disease (PD). Common examples include spousal infidelity, abandonment, intruders near or in the home, suspected stealing and other somatic and grandiose ideas [1]. Delusional psychosis affects about 16% of the PD population [2]. To date, much of existing research examines the biological and pharmacologic underpinnings of psychotic phenomena in PD [3]. For instance, a review of dopamine agonists’ association with delusional thinking, paranoia and jealousy, hallucinations and problems of impulse control such as hyper-sexuality, gambling and grandiose ideas revealed a major impact on the quality of life for the person with PD (PWP) [4]. A systematic review of published PD cases related to delusions examined the association between disease progression and cognitive impairment with delusions [5]. Authors found that delusions were often paranoid, associated with earlier onset of PD and higher rates of impulse control disorders and dopamine dysregulation syndrome, yet lower rates of cognitive impairment. Thus, they concluded that delusional psychosis is not adequately explained by existing models of PD psychosis.

Enhanced focus on psychosocial topics in PD has been quantitative [6,7]. These studies document increased caregiver burden with psychosis, often associated with dementia in PD. Qualitative research can provide nuance and narrative to quantitative surveys or evaluations. A qualitative metasynthesis found challenges for family carers, including maintaining their own health, wanting knowledge, but also fearing it (“perhaps it is better not to know”) [8]. Specific research on spouse carers involved a range of topics, including “middle aged” spouses, the effect on the spousal relationship, living with a partner with PD psychosis, how spouses perceive their caregiving experiences and the healthcare system [9,10,11,12,13,14,15,16,17,18].

Studies specific to delusions include a systematic review extracting the results of psychosis studies and a study of the PD patients’ experience of psychosis [19]. The latter highlighted four themes: emotional aspects of delusions for PWP; the sense of uncertainty and losing control; loss of identity and sense of self; acceptance and adjustment to the experience of delusions. A growing body of qualitative research explored how partners and caregivers of PWP were affected by their loved ones’ PD [20,21,22,23,24,25,26,27]. A study of the caregiving experience for someone with PD psychosis found that coping could be improved if carers had assistance to process feelings associated with their loved ones’ PD psychosis. A recent study found that care partners of PWP and psychosis often discussed burden and guilt, triggers to psychosis and strategies to care for loved ones with psychosis [28]. Three participants in this study experienced jealous delusions. Challenging emotions and consequences included anger, lowered mood, stress, loss of social networks and diminished self-esteem [28]. Other work documented the mental health and relational problems for carers of psychotic persons, not specifically PD [29].

The existing literature regarding neurobehavioral problems in neurology patients often surrounds dementia and, in particular, Alzheimer’s Disease (AD) [30,31,32]. Psychosis in AD does not have the added complexity of motor impairment dependent on treatment that can exacerbate psychosis. Further, treatment of psychotic symptoms in AD involves some similar general principles to PD, including a search for exacerbating factors (new medications, non-adherence to medications, infections, general medical problems, pain, depression) but recommended treatments involve the use of tranquilizers such as Risperidone that are contraindicated in PD [33]. PD psychosis, including delusions, requires balancing the motor function and neurobehavioral symptoms and may initially be managed by only adjusting PD medications.

Given that the content of PD jealous delusions is often intense and intimate (e.g., delusions around marital infidelity), PD delusions are uniquely threatening to marital relationships in the face of pre-existing caregiver strain. What is normally personal and private may be aired in public by the PWP when delusional, adding additional strain and suffering for the spouse carer. The implication of disclosing delusional thoughts has ramifications for the spouse carer and the PWP. The previous literature has combined psychotic symptoms when considering caregiver burden and thus, have not given voice to the lived experience of spouse caregivers subjected to jealous delusions. Further, clinicians’ understanding of psychosis impact was largely drawn from the quantitative literature, ignoring the personal and individual experience of jealous delusions. This has resulted in clinicians focusing on the biological phenomena of psychosis as a single phenomenon and ignored the impact on spouse carers. When the impact on caregivers was discussed in the literature at all, the focus was on grief and increased burden. Based on our experience of carer burden associated with PD delusions in a Neuropalliative care clinic, we sought to use qualitative research to document the lived experience of spouse carers of PWP and jealous delusional phenomena. Using qualitative methodology, we asked: How were PD delusional phenomena understood by spouse carers? What was their psychological effect on spouse carers? How did spouse carers cope with PD delusions?

## 2. Methods

### 2.1. Research Participants

This study was approved by the Research Ethics Board of the University of Alberta, Pro00070834. Volunteers were recruited from a tertiary Parkinson and Movement Disorders Program at the University of Alberta. The participants were identified by the Neuropsychiatrist, Neurologist or Nursing staff who were aware of PD jealous delusions. The inclusion criteria were: partner of a PWP with active or recent past delusional thoughts; no age restriction; male or female; fluent in English and able to consent to research.

### 2.2. Qualitative Interviews and Analysis

Individual, semi-structured 30–60 min interviews were audiotaped (without the PWP being present) in person or over the phone. Audiotapes were transcribed by a professional research transcriber who signed a confidentiality agreement. Transcribed interviews contained words of both interviewer and participants. Each script was identified as a subject number (e.g., SP 1, SP 2). Baseline data included: the age of the participant and PD patient, PD duration. The interview focused on examples or evidence of delusional thinking, their awareness of the possibility of PD delusions prior to onset, what was helpful for coping, the participant’s own sense of their personality, what could be helpful for caregiver spouses in similar situations and a chance to ask any questions that they had resulted from the interview process (See Appendix A).

Transcriptions were coded using MAXQDA 2018, a qualitative analysis software package. Based on this analysis, themes were identified and coded until saturation, and an analysis of emergent themes was carried out to inform understanding of spousal experiences of delusions in PD. To optimize qualitative validity and qualitative reliability, the following steps were taken: (1) reading and re-reading the transcripts to ensure a full understanding; (2) ensuring a clear definition of each code was established and examples provided (included in memo sections within MAXQDA 2018); (3) numerous meetings to discuss with the second coder to discuss coding, findings and analysis of findings; (4) cross-check coding with two other researchers to assess and compare results; (5) engagement of external readers without the familiarity of the topic for independent review; (6) codes that emerged were then aggregated, and the findings were grouped and presented as thematic summaries in keeping with the guidelines for quality qualitative research; clarification of code definitions and review to avoid a shift in meaning or analysis drift.

### 2.3. SPs Withdrawal from Research

SPs could withdraw up to two weeks after being interviewed. On the information letter that accompanied the informed consent form, participants were told that the data would be published in a healthcare journal, and the article would be mailed out to them if they wished to have a copy.

### 2.4. Data Availability

The data are available to qualified researchers upon request to CD. Data available will be transcribed interviews. This is not publicly available since details of the interview may identify participants.

## 3. Results

### 3.1. Participant and PD Patient Characteristics

Twelve spouse participants were recruited. Four men and eight women with an age range from 66 to 80 years participated (Table 1). The duration of PD for their spouse was 3–24 years (mean 13.6 years). The saturation of themes was achieved at participant 9. Comments from all 12 participants were coded. No subjects withdrew from the study.

### 3.2. Delusion Descriptions

Jealous delusions were experienced by all SPs. These surrounded infidelities, bringing someone home to have a romantic relationship or accusations of the SP being a prostitute. Jealous delusions often occurred in PWP who had delusions surrounding other people. If delusions involved other individuals, the SPs were often able to realize thoughts were not based on reality.


*SP 3: “He very conspiratorially would whisper to me that they were running a prostitution ring there and that all the nursing staff was in on it. I realized, well that’s not adding up at all!”*



*SP 3: “I thought we were having a really good visit and he whispered to me to come closer so he could tell me something and I thought he was going to tell me something wonderful like he loved me or thank-you. And he whispered, ‘Just so you know, everybody here thinks you’re a hooker.’”*


Jealous delusions were no less divorced from reality but caused SPs to be hurt by these accusations. SPs often tried to reason with the PWP to no avail. Often SPs described spending much of their day accompanying or helping the PWP leaving little time to engage in activities that they were accused of.


*SP 6: “First I said, ‘Don’t be silly’, and pointed out that we were together every day, all day, aside from maybe 15 min that I’d watch the news. I did that because it was a busy time of the night getting my wife ready for bed and that was my 15 min of collapse.”*



*SP 8: “I spent a lot of time in town and when I came back home she says ‘Okay, who’s that woman you’re going with?’ … I laughed it off.”*


Persistent, jealous delusions inevitably had an impact on the SPs and their relationship with the PWP leading to strategies for SPs to cope.

### 3.3. Relationship Changes

As a result of PD delusions, SPs described profound changes in their marital relationship.


*SP 3: “I was extremely sad that he was gone, that the essence of him was starting to disappear. This wasn’t the person that I had known. We’d always been attached at the hip and he was always the go-to person for me to talk to about my concerns with a variety of things…but I couldn’t talk to him about that.”*


In addition to taking steps to avoid triggering psychotic thoughts, SPs felt they had lost their life partner, who was often their primary source of advice and psychological support. SPs continued to love their spouse and acknowledged the change in the relationship.


*SP 3: “I really love him very much, but as far as a partner completely in marriage, he is not there for sure anymore.”*



*SP 6: “Her delusions changed our life … she would talk about me not realizing it was me she was talking about.”*



*SP 12: “You can’t be the primary caregiver and be vilified at the same time. It’s really a quite impossible task.”*



*SP 9: “Trapped in my own house, trapped everywhere.”*


Given the jealous delusions and their untrue nature, SPs described struggling with ongoing care for the PWP. SPs disclosed how they coped with the hurt of the present by recalling their past relationship, and for many, the person before delusions became a daily struggle.


*SP 5: “You’ve got to remember how much you loved this person because that’s who they really are.”*


### 3.4. Spouse Strategies for Jealous Delusions

#### 3.4.1. Managing Incredulity


*SP 7: “Just puzzled why she would think that. There were other comments about when I was going to leave because of the stress of looking after her and she would say things that I hadn’t said about leaving her”.*



*SP 3: “At first it was, “How dare you! What did I do to deserve this?” And then it’s like, “OK, we’ve got to remember he’s not well. This isn’t normal. Let’s talk about it in the morning and see what happens”.*



*SP 2: “He always thought I was a party girl for some reason. I wasn’t but he assumed for some reason that I was always out with somebody and being really rotten and unfaithful. This went on and on and that was very, very hurtful. We constantly lived together, so I can’t imagine when I would have had the time to do this, but that’s what a lot of his thoughts were”.*



*SP 1: “It catches you off guard because it’s nothing and then there it is. It starts. I didn’t have any sense that that was going to start happening.”*


Most SPs reported the delusions of the PWP as surprising and shocking and an unexpected facet of PD. The sudden onset of delusions upset the homeostasis of the marriage, and the content was unimaginable and hurtful (marital infidelity, being replaced by an imposter, being a prostitute and running a prostitution ring, etc.). PWPs’ opinions, statements and responses were initially believed by SPs when involving others (someone has stolen from them, is poisoning them, wishes them ill, etc.). However, when delusion content pertained to the SPs, it was reported as a personal affront, especially if care needs related to PD was particularly burdensome.

SPs reported that they eventually understood that delusions resulted in their partners saying and doing things not based on reality. Despite this knowledge, the delusions created doubt and uncertainty regarding the entirety of their marriage. SPs coped by focusing on the duration of their relationship and past challenges successfully overcome together in order to reduce pain and isolation. Some participants considered leaving the marriage. While attempting to make sense of PD delusions, SPs ruminated over what their partners said. Despite lack of evidence to support delusional thoughts, participants began questioning events, words and gestures that might have provoked delusion content. This resulted in constant questioning of their own and PWPs’ words and actions for hidden meaning.

SPs described the PWPs’ personalities as changed by delusions. As a result, SPs sought the partner they had once committed to and attempted to rationalize and understand the “new” spouse’s words and actions. Some SPs focused on who their spouses were before the onset of PD delusions to reduce their pain and maintain the marital bond. As an example, SPs would frequently recall past times when their partners were loving and kind. SPs made excuses for the PWP in an effort to protect their partners and perhaps to reduce their own suffering. Although mindful of the present, SPs used memories to extract happy recollections of when times were good. Those memories helped to sustain them so they could continue to look after their partners and to remain dedicated and faithful spouses.

#### 3.4.2. Hypervigilance


*SP 9: “There was a paranoia, a feeling that he was going to be potentially violent. It was terrifying because I didn’t know what he was going to do to me. I could picture him coming into my bedroom with scissors in his hand and killing me. That’s how terrifying he was. He never did, and he never came at me with anything, but it was that scary to me”.*



*SP 12: “I had to be really careful that I didn’t trigger something that would upset him because he would become very upset”.*



*SP 3: “I’m just putting my guard up and waiting and trying to be as calm as I can with the situation*
*”.*


SPs described hypervigilance. The onset of PD delusions often resulted in volatile behavior that threatened the SPs’ sense of well-being and safety. They were perpetually preparing for the next attack or accusation. SPs described a constant state of readiness with little mental rest and, despite this level of alertness, were repeatedly caught off guard. Sub-themes of danger and violence were raised frequently, with SPs emphasizing the pervasive potential for their delusional partners’ outbursts.


*SP 4: “It was happening so quickly that honestly I didn’t have much time to think about myself. I was just acting on what was coming because it was one thing after another. He saw the house on fire. Then he started thinking we were flooding. Then he ran away from the house.”*



*SP 3: “I know it’s probably going to happen again. So, let’s see what we can do to avoid or stay on top.”*


As part of hypervigilance, SPs reported profound changes to their routines and activities. This included monitoring words and actions or ensuring that absences from the home could be corroborated. They reported greater self-consciousness of their own behavior and expressions, mindful that apparently innocuous actions might provoke partners’ suspicion or paranoia. Even recalling and talking about their experiences to the interviewer was difficult. Despite the state of hypervigilance, participants consciously adopted a calm exterior for the benefit of the PWP and the outside world.


*SP 9: “It’s got me nervous, upset, just thinking about it again.”*



*SP 4: “The good thing about this was that I learned that I’m capable of physically restraining him…Of course, I learned what to do, but there was a moment that I didn’t brace myself properly, so he was able to shake me.”*


The PWPs’ delusions caused SPs to be physically, mentally and emotionally exhausted and contributed to transforming the spousal relationship to work. When healthcare providers did acknowledge SPs’ exhaustion, the proposed solution was time away from the PWP. Although well-meaning, this advice posed a new set of problems for the SPs. Time away from the PWP required extraordinary measures to avoid provoking paranoid thoughts and behaviors. One SP had to hide her car since her spouse searched for her when she spent one night at a friend’s home. Therefore, SPs were caught in a dilemma of needing respite, but knowing this might trigger delusional thoughts and therefore, actions. SPs reported the dual tasks of protecting themselves from violence from the PWP and protecting their spouse from harm that may have resulted from acting on their delusions.

#### 3.4.3. Defensive Strategizing


*SP 9: “I was being sneaky. Only because that was the only way I could contact my family. I ended up hiding the phones on him every night when I went to bed. All the phones I had to hide. I had to hide the car keys because I didn’t know where he was going to go if he went out. It was a holy terror. I just can’t explain how bad it was”.*



*SP 6: “Living with her delusional, I will be direct with her but if she is confrontational, I won’t continue. I’ve to learn not to over control the situations but I do need to manage her daily living because she will put the milk in the microwave and the cheese in the freezer”.*



*SP 3: “You’ve got to pick your battles and you’ve got to let go”.*


The theme of defensive strategizing captured SPs’ creative strategies to manage their delusional partners. SPs had to make decisions without the help of their spouse, which was often a drastic change from previous decision-making for the couple. SPs assumed the role of primary decision-makers and accepted the limitations of the PWP often putting their own needs last if considered at all. This had to be managed carefully to avoid accusations of wishing the PWP harm or being part of a plot to harm the PWP.


*SP 1: “As long as I can control where he’s at. I can control the people he’s with. I work very hard to make them understand what changes have happened, so they are watching for them, so that he doesn’t get into any kind of trouble.”*


Another defensive strategy was to actively adopt a positive outlook despite the marital relationship challenges. The SPs reported positivity helped them cope with chaos from the PWP’s delusional thoughts and resulting actions. SPs felt societal pressure to put forward the image of competence and full commitment to their spouse in front of health professionals, friends and family. For some SPs, this meant concealing delusions or the full effect of delusional thoughts on their own mental and physical health. SP’s inability to disclose their suffering resulted in fewer interventions from healthcare professionals and less support from family members.

#### 3.4.4. Concealing and Exposing


*SP 11: “Well, I don’t want to be a burden to those that are close to me so that they stop being as close as they are to me. So many friends have gone by the wayside through all of this and I just want to go out and laugh again. I want to have ‘a good belly laugh’ as they say. I want to wake up and not feel the weight of the world . . . because people don’t want to be around somebody who is sad all the time”.*



*SP 12: “That’s just normal—to put on the best face. That’s what you do and then you expect him to do that too”.*



*SP 2: “It’s just something I guess at my age you just don’t speak up and discuss it. We weren’t used to discussing that kind of stuff in public . . . sexual things out in public. Not like now. You see it on TV and it’s very common”.*


SPs reported the conflict between complete disclosure to healthcare providers and concealing the truth. SPs wanted to maintain the dignity of their spouse and be loyal. “Guilt” was a common word associated with delusion disclosure. This led to SPs not disclosing the PWP’s delusional thoughts to adult children to maintain their spouse’s dignity or standing in the family. SPs often did this at the expense of their own family or friend relationships. They rarely discussed their concerns for their own safety to family or healthcare professionals. Concealing delusions may have been related to the delusion content (sex, prostitution and infidelity or mistreatment). SPs were uncomfortable speaking about sexual content—even though the delusions were not true. The sexual content of delusions added to carers’ distress.

The desire to conceal delusions led to social isolation as SPs could not predict when the PWP would disclose their delusions. Therefore, respite for social connection or family events was diminished for the PWP but also the SPs.

SPs did not disclose delusions to their own Primary Care Providers despite the impact on their own lives. Psychologically, many SPs might have compartmentalized delusions to cope and thus, did not disclose delusions to the healthcare providers who could help them.


*SP 3: “I don’t even know if I talked about it to the family doctor…and yet her father had Parkinson’s.”*


SPs acknowledged their role as caregivers but also reported their own needs for support and validation. Given the frequent nondisclosure, carers were frequently not able to have their own needs met. Participants reported conflicting emotions of needing to appear competent to care for their spouse and handle everything, yet experiencing shame, guilt and sadness. Potential support from family or healthcare providers was often denied to SPs as they delayed disclosure or minimized impacts. Therefore, the conflict between concealing and disclosing may have contributed to feelings of hopelessness, isolation or being trapped.

## 4. Discussion

The analysis and synthesis of interview data elicited themes capturing the psychological effect for those interviewed of living with a partner who has PD delusions. Spouses were adversely affected psychologically by their partners’ delusions. They reported shock and emotional pain as they tried to comprehend bizarre and attacking comments. SPs adopted hypervigilance for their own safety and that of their partners. They reported defensive strategizing, which encompassed blunting their emotions to cope with unexpected fury from their changed spouse. Even mundane tasks required incredible machinations to account for absences from the home. Despite their humiliation by spouses’ disclosing delusions to others, they attempted to be competent as a caregiver and decision-maker and protector of their spouses’ dignity. This created a tension between their public persona of the committed carer while feeling emotionally and, sometimes, physically fearful at home. SPs nondisclosure of spouse delusions may have resulted in a delay in diagnosis and interventions and hence, more trauma for both the SP and PWP. Further, nondisclosure to family and healthcare providers may reflect a state of SP denial. If prolonged, denial could result in more internal psychological conflict for the SP as well as the inability to acknowledge the loss that PWP delusions represent to the marital relationship.

Healthcare research tended to focus on patient experiences of delusions to the exclusions of the caregiver experiences of delusions in PD. While the literature on related topics exists, such as the effects of gender differences in sexual behaviors of Alzheimer disease or intimacy issues due to dementia [34,35] and the high prevalence of physical and sexual aggression towards caregivers in advanced PD and experiences in palliative care [36], there is no specific body of research addressing delusions from the caregivers’ perspective in a rich, qualitative manner. Our findings suggest changes in, and major challenges to, the psychological health of spouses of PWP who have delusions. Experiencing violence and emotional abuse is not uncommon for caregivers looking after PWP. Intimate partner violence was recorded in the general body of the research literature pertaining to marriage [37,38,39], as have topics that address ways of coping when loved ones have dementia [40]. The common element, as suggested by the findings, is the suffering of spouses due to the illness or behavior of their partners. SPs in our study revealed deeply intimate and personal changes that were rarely discussed outside the couple unit, if at all. SPs experienced a breach in the spousal relationship due to the PD delusions yet attempted to remain faithful and helpful as life partners and the primary caregiver despite their often silent suffering.

Relational changes as a result of the PWP delusions were profound. Changes included loss of trust between the spouses, intimacy and, at times, love. SPs described a loss of marital mutuality and change to the caregiver that they had not anticipated. Many SPs reported continuing in the caregiving role to honor their marital commitments. In addition to psychological distress, SPs reported social isolation, loss of outside relationships and their own physical challenges due to lack of sleep and heightened anxiety.

SPs often reported shame and embarrassment regarding their spouse’s delusions. SPs felt guilty about disclosing delusions since it may reflect poorly on their spouse. Responses often reflected minimizing delusions or attempting to counterpoint delusions with prior “good” behavior. All SPs reported that the interview was the first time they had a focused discussion about the delusions and consequences for them. It would have also been helpful to ask or learn more about what their adult children experienced as well. Was the experience the same or different? Adult children, too, are silent voices in the literature regarding how PD delusions affect the family unit. The input of other caregivers may also have been a good resource for a thorough examination of the research question.

Based on our work, spouses and their partners should be exposed to the topic of delusions. *(“No one told us about delusions, I think they should have warned us”*). However, timing this information is challenging. (*“not everybody wants to know”*). Encouraging spouses or family members to attend clinical appointments can allow for consistent information sharing between health professionals and patients and family. Framing information in a neutral manner that delusions (or hallucinations) can occur in some people with PD may be an acceptable manner to disclose this possibility of their occurrence to PWP and spouse care partners. Screening for delusions should occur at least annually for those with PD [41].

Education specifically for care partners of PWP can provide the spectrum of neurobehavioral changes that might occur in PWP and that for many, there are opportunities to improve neurobehavioral changes. Earlier identification of delusions may reduce the burden for care partners and avoid relationship changes our respondents experienced. Generically, care partner interventions that teach Cognitive Behavioral Therapy techniques and problem-solving abilities can improve psychosocial outcomes, self-efficacy and physical health for caregivers. Care partners of PWP with delusions may require additional support either through their own primary care provider or mental health professional. While the neurologic workforce at this time is limited, attention to care partner health may have an impact on our PD patients’ outcomes [42,43]. Therefore, with multidisciplinary care, we may be better able to address both care partners’ and PWPs’ needs.

The strengths of this study include the qualitative approach. This allowed exploration of the impact of delusions for spouse carers through their lived experience. The identified themes can serve to inform educational offerings for healthcare providers in addition to interventions for those spouse carers of PWP with delusions. Although not included in our study, we propose that family caregivers may also experience similar trauma, anxiety and conflict when faced with a family member with delusions.

The limitations of our study are that SPs were interviewed once for 30–60 min. For many, this was the first meeting with the interviewer. Due to the sensitive nature of the information disclosed, this may have hindered open conversation. Multiple interviews might have increased thematic content; however, many SPs felt very strained to find even 30–60 min of private time. Some interviews were conducted over the phone for participants’ convenience. There was no evidence within the data that the face-to-face interview data differed from that retrieved through telephone interviews, but the setting of each may have hindered more revealing insights. In fact, telephone interviews may have allowed participants a sense of true anonymity to increase the ease of disclosure. For those interviewed in person, an interview room in the same area as the PWP clinic visits was used. This may have been less conducive to full disclosure by the SPs, although, for others, the familiar setting may have increased disclosure.

Future research on carers of those with PD should include the impact of neurobehavioral changes associated with PD. Interventions to relieve psychological complications from neurobehavioral changes of their spouse or family member can help maintain relationships and allow the PWP to remain at home as long as possible. Education of spouse and family carers regarding neurobehavioral changes should occur at any stage of illness and should be framed to convey that the majority of PWP never experience these changes. By providing information to those who are best placed to detect such changes, we can help PWP and their carers live high-quality lives.

## 5. Conclusions

Jealous delusions have profound impact on spouse carers and their relationship with the PWP and others (family, healthcare providers, friends, etc.). Spouse carers reported many strategies to cope with jealous delusions including defensive strategizing, hypervigilance, managing credulity and the conflict between conceling and exposing the PWP’s jealous delusions. While some of these strategies helped them cope, they could also result in delay of identification of jealous delusions to healthcare providers and hence, intervention to resolve or minimize them. Neurologists can improve detection of delusions by regularly screening for them in a non-judgmental manner. Informing spouse carers and other involved in the care of PWP of the possibility of delusions can help them identify delusions as a complication of PD that should be discussed with their neurology team. Our work highlights that delusions are not just a neurobiologic phenomena but a potential cause of harm and burden to spouse carers of PWP.

## Figures and Tables

**Table 1 brainsci-11-00871-t001:** Participants’ demographic data.

Participant #	Gender	Age	Years Since PD Diagnosis
1	F	73	8
2	F	74	23
3	F	62	18
4	F	74	16
5	F	74	8
6	M	66	13
7	M	80	10
8	M	79	3
9	F	80	6
10	M	82	24
11	F	62	16
12	F	70	18

## Data Availability

Original qualitative de-identified interviews are available to qualified researchers upon request to the corresponding author.

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
