# Peer review of "Psychological Impact of Parkinson Disease Delusions on Spouse Caregivers: A Qualitative Study"

_brainsci, 2021, doi:10.3390/brainsci11070871_

Round 1

Reviewer 1 Report

I think this is an important paper as it provides a useful insight into how Parkinson's disease delusions affect spouse caregivers. I have the following suggestions for the authors.

The background section/introduction should be rewritten to provide a clear rationale for the study. The first paragraph on page 2 is hard to follow as it moves from talking about quantitative research to qualitative and (I think) back again but I was not sure.

I think the rationale for the study is that there is a lack of research into how spouse caregivers experience delusions associated with PD, yet the authors quote a lot of existing literature – can you clarify how this work contributes to knowledge - one argument I would add in is that PD is different to AD and most of the current advice and guidance for caregivers is related to AD care.

I thought the methods were clearly defined but I was left with the question of whether any participants did withdraw.

There is extensive discussion in the results section, which appears to me to be based on quite thin data (eg lines 154-163 under the first theme). There are not many quotes and no more in the supplementary data provided (also not sure why the quotes are in the supplementary data when no additional data are included). Please make sure that all your commentary is evidence based. All the discussion should be supported by transparent data – ie quotes.

Additional material is also presented in the discussion that was not in the results – eg describing the caregiving as “job-like”. If this is a finding, it should be in the results section, evidenced by the direct quote and then referred back to in the discussion section.

Your recommendations for health professionals are useful – you could expand on how this could be put into practice.

Author Response

I think this is an important paper as it provides a useful insight into how Parkinson's disease delusions affect spouse caregivers. I have the following suggestions for the authors.

The background section/introduction should be rewritten to provide a clear rationale for the study. The first paragraph on page 2 is hard to follow as it moves from talking about quantitative research to qualitative and (I think) back again but I was not sure.

Response:  We thank Reviewer #1 for their comments.  We only mentioned quantitative research on psychosocial topics.  We have made a more fullsome discussion to increase clarity. 

I think the rationale for the study is that there is a lack of research into how spouse caregivers experience delusions associated with PD, yet the authors quote a lot of existing literature – can you clarify how this work contributes to knowledge - one argument I would add in is that PD is different to AD and most of the current advice and guidance for caregivers is related to AD care.

Response:  We have added citations regarding dementia psychosis and discussed the differences in approach to management to highlight differences with PD psychosis.  

Existing literature regarding neurobehavioral problems in neurology patients often surround dementia and in particular Alzheimer Disease  (AD) (cite).  Psychosis in AD does not have the added complexity of motor impairment dependent on treatment that can exacerbate psychosis.  Further, treatment of psychotic symptoms in AD involve some similar general principles to PD including search for exacerbating factors (new medications, nonadherence to medications, infections, general medical problems, pain, depression) but recommended treatments involve the use of tranquilizers such as risperdone that are contraindicated in PD (cite).  PD psychosis including delusions requires balancing the motor function and neurobehavioral symptoms and may initially be managed by only adjusting PD medications in the introduction highlighted. 

I thought the methods were clearly defined but I was left with the question of whether any participants did withdraw.

Response:  No participants withdrew.  This has been included in the Results.  

There is extensive discussion in the results section, which appears to me to be based on quite thin data (eg lines 154-163 under the first theme). There are not many quotes and no more in the supplementary data provided (also not sure why the quotes are in the supplementary data when no additional data are included). Please make sure that all your commentary is evidence based. All the discussion should be supported by transparent data – ie quotes.

Response:  We have increased the quotes from interviews were appropriate to increase the validity of conclusions.  We have also removed the Supplementary table of quotes as, Reviewer #1 is correct, it did not add to the manuscript.  

Additional material is also presented in the discussion that was not in the results – eg describing the caregiving as “job-like”. If this is a finding, it should be in the results section, evidenced by the direct quote and then referred back to in the discussion section.

Response:  We have added a section in Results:  Relationship Changes.  This houses the quotes that support statements included in the Discussion.  We thank Reviewer #1 for highlighting this error.  

Your recommendations for health professionals are useful – you could expand on how this could be put into practice.

Response:  We have increased the discussion of how to put our recommendations in place for health professionals including CBT and problem-solving skill development for caregivers.  

Reviewer 2 Report

Deutsch et al. have conducted a qualitative analysis of care-partner emotional responses to delusional psychosis in their spouse with PD. This is an important topic, and the detailed analysis, together with quoted examples of care-partner emotional responses, is a critical addition to the literature. I have a few minor comments/suggestions to consider in revision:

  1. The introduction/literature review is missing some relevant papers on caregiver burden related to psychosis specifically and advanced disease generally, including Martinez-Martin et al 2015 (10.1016/j.parkreldis.2015.03.024.), Champagne and Muise 2021 (10.1177/0033294121998032), Mantri et al 2021 (10.1371/journal.pone.0248968), Rosqvist et al 2021 (10.1155/2021/9475026).
  2. Was the content of the delusion assessed? I imagine that delusions of grandeur would have a very different impact on the care-partner than a Capgras delusion or delusional jealousy.
  3. Was there any discussion of the impact of medications (either medications to treat delusions or medications that may precipitate it such as anticholinergics or dopamine agonists)?
  4. The experience of neuropsychiatric symptoms, both by patient and care-partner, is colored by cultural background. Was any data collected on race/ethnicity/cultural identification of either the care-partner participant or the spouse with PD?
  5. The use of pseudonyms, as opposed to subject identifiers, in qualitative research is controversial (see Guenther 2009 https://doi.org/10.1177/1468794109337872; Allen and Wiles 2016 https://doi.org/10.1080/14780887.2015.1133746), particularly if pseudonyms are assigned by researchers rather than chosen by participants. As this study is cross-sectional rather than longitudinal, and as the presented quotations are relatively short, I would suggest the authors use subject IDs instead of renaming their participants -- the slight loss of intimacy to the reader is better than the impact of a participant reading their words assigned to a different identity.

Author Response

We thank Reviewer #2 for their helpful comments and links for references. 

  1. The introduction/literature review is missing some relevant papers on caregiver burden related to psychosis specifically and advanced disease generally, including Martinez-Martin et al 2015 (10.1016/j.parkreldis.2015.03.024.), Champagne and Muise 2021 (10.1177/0033294121998032), Mantri et al 2021 (10.1371/journal.pone.0248968), Rosqvist et al 2021 (10.1155/2021/9475026).

Response:  We added Martinez-Martin and Mantri that specifically mention psychosis and delusions.  Champagne and Muise and Rosqvist et al, although excellent manuscripts, did not mention the impact of psychosis or delusions in their work and therefore, we have not included them.  

  1. Was the content of the delusion assessed? I imagine that delusions of grandeur would have a very different impact on the care-partner than a Capgras delusion or delusional jealousy.

Response:   All PWP had jealous delusions surrounding the care partner.  We have included a section Delusion Content to clarify this important question and included representative quotes regarding delusion content.  

  1. Was there any discussion of the impact of medications (either medications to treat delusions or medications that may precipitate it such as anticholinergics or dopamine agonists)?

Response:  As this was a study of the impact on spouse caregivers, we did not examine the biologic basis of the individual’s psychosis. 

  1. The experience of neuropsychiatric symptoms, both by patient and care-partner, is colored by cultural background. Was any data collected on race/ethnicity/cultural identification of either the care-partner participant or the spouse with PD?

Response: 

We did not collect race or ethnicity data on the subjects although have data to suggest that our clinic provides care to 90% white Albertans.  We regret not collecting this data originally.   We have added this in limitations.  

  1. The use of pseudonyms, as opposed to subject identifiers, in qualitative research is controversial (see Guenther 2009 https://doi.org/10.1177/1468794109337872; Allen and Wiles 2016 https://doi.org/10.1080/14780887.2015.1133746), particularly if pseudonyms are assigned by researchers rather than chosen by participants. As this study is cross-sectional rather than longitudinal, and as the presented quotations are relatively short, I would suggest the authors use subject IDs instead of renaming their participants -- the slight loss of intimacy to the reader is better than the impact of a participant reading their words assigned to a different identity.

Response:  We agree with Reviewer #2 and have changed quotes to be attributed to the SP #, removed pseudonyms from the chart and removed reference to pseudonyms from the Methods section.

Round 2

Reviewer 1 Report

> The authors have amended the manuscript according to my feedback in many areas and the findings are now much stronger. However, the introduction still lacks a clear argument as to how this study fits within the broader scholarship in this area - the introduction should end with the question that they are addressing, which should clearly flow from a discussion of what is already known and what the gaps are. I believe this needs further work prior to publication.

Author Response

We thank Reviewer 1 for their helpful suggestions.  We have made the suggested changes to better outline why our research fills a gap in psychosis research.  In particular, our research focuses on jealous delusions that are integrally threatening to the marital relationship and yet, not uncommon in practice.  

Given that the content of PD jealous delusions is often intense and intimate (e.g., delusions around marital infidelity), PD delusions are uniquely threatening to marital relationships in the face of pre-existing caregiver strain.   What is normally personal and private may be aired in public by the PWP when delusional adding additional strain and suffering for the spouse carer. The implication of disclosing delusional thoughts has ramifications for the spouse carer and the PWP. Previous literature has combined psychotic symptoms when considering caregiver burden and thus, have not given voice to the lived experience of a spouse caregivers subjected to jealous delusions.  Further, clinicians’ understanding of psychosis impact has largely been drawn from quantitative literature, ignoring the personal and individual experience of jealous delusions.   This has resulted in clinicians focusing on the biologic phenomena of psychosis as a single phenomenon and ignored the impact on spouse carers.  When the impact on caregivers was discussed in literature at all, the focus was on grief and increased burden.  Based on our experience of carer burden associated with PD delusions in a Neuropalliative care clinic, we sought to use qualitative research to document the lived experience of spouse carers of PWP and jealous delusional phenomena. Using qualitative methodology, we asked: How were PD delusional phenomena  understood by spouse carers?, What was their psychological effect on spouse carers?  and How did spouse carers cope with PD delusions?